# Progesterone Promotes Anti-Anxiety/Depressant-like Behavior and Trophic Actions of BDNF in the Hippocampus of Female Nuclear Progesterone Receptor, but Not 5α-Reductase, Knockout Mice

**DOI:** 10.3390/ijms26031173

**Published:** 2025-01-29

**Authors:** Cheryl A. Frye, Daina M. Cleveland, Anjali Sadarangani, Jennifer K. Torgersen

**Affiliations:** 1Comprehensive Neuropsychological Services, 490 Western Avenue, Albany, NY 12203, USA; dainac2004@gmail.com (D.M.C.); anjalis13@berkeley.edu (A.S.); torgersen.jennifer@gmail.com (J.K.T.); 2Department of Psychology, The University at Albany, 1400 Washington Avenue, Albany, NY 12222, USA

**Keywords:** elevated plus maze, forced swim, open field, neurosteroid, neuroactive steroid, non-genomic

## Abstract

Progestogens’ anti-anxiety and anti-depressive effects and mechanisms are not well-understood. Progestogens are hypothesized to have anti-anxiety and anti-depressive effects on behavior, independent of actions at nuclear progestin receptors (NPRs) and dependent on allopregnanolone (5α-pregnan-3α-ol-20-one; 3α,5α-THP), a 5α-reduced, neuroactive metabolite of progesterone (P_4_). Adult c57 mice in behavioral estrus (proestrus; pro) showed more anti-anxiety-like and anti-depressant-like behavior and higher levels of estradiol (E_2_), P_4_, and allopregnanolone in the hippocampus/amygdala complex. Proestrus c57 > 5α-reductase knockout (5αRKO) mice made more central entries in an open field than diestrus c57 and 5αRKO mice that were not different. Ovariectomized (OVX) c57 mice administered 1, 2, or 4 mg/kg P_4_ SC showed dosage-dependent increases in central entries in an open field (more anti-anxiety-like behavior); 5αRKO mice had maximal increases at 1–2 mg/kg P_4_. OVX c57 and 5αRKO mice showed maximum increases in central entries with SC 3α,5α-THP (4 mg/kg), and c57s showed a similar maximal response to P_4_ (4 mg/kg), but 5αRKOs response was half at that dosage. P_4_ (4 mg/kg SC to OVX c57 or progestin receptor knockout (PRKO) mice decreased immobility (depression-like behavior) in the forced swim task. Effects of E_2_ and veh were similar in both groups. Levels of 3α,5α-THP in the hippocampus/amygdala were consistent with effects on central entries in the open field. Levels of brain-derived neurotrophic factor (BDNF) in the hippocampus/amygdala were greater among E_2_-primed (0.09 mg/kg, SC) vs vehicle-administered mice. In sum, adult female mice can be responsive to P_4_ for anti-anxiety/anti-depressant-like behavior; such effects may be independent of NPRs but require 5α-reduction and E_2_’s priming actions at BDNF in the hippocampus/amygdala complex.

## 1. Introduction

Progesterone (P_4_) plays a key role in reproduction, cyclicity, onset, and maintenance of pregnancy, as well as many other trophic actions throughout the life span. Among females, a primary source of P_4_ is the corpus luteum, which forms from ruptured follicles, where ova are released, and produces P_4_ until the placenta takes over, or the regression of the corpus luteum occurs as the menstrual cycle starts anew. P_4_ can also be synthesized by adrenals, placenta, and brain [1,2]. P_4_ is produced in smaller amounts by adrenals, testes, and the brain; it plays a key role in sperm production, sexual function, and trophic action [3]. The essential functions or key physiological effects of P_4_ occur through various mechanisms of action. A comprehensive review of the two most studied actions of P_4_ is beyond the scope of this paper. One key mechanism by which P_4_ functions is by binding to nuclear P_4_ receptors (NPRs): P_4_ binds to specific NPRS in target tissues, such as the uterus, breast, and brain. Once bound, NPR complexes act as transcription factors, modulating gene expression and influencing cellular processes. NPRs, upon binding to P_4_, adhere to specific DNA sequences called P_4_ response elements within the promoter regions of target genes. This binding can either enhance or suppress gene transcription, leading to the synthesis of proteins that mediate P_4_’s effects. PRKO mice, which have nonfunctional NPRS, are utilized in some of the experiments discussed herein [4].

Dr. Bert O’Malley has led investigations of P_4_’s mechanisms of action through NPRs, which are comprehensively described in a review paper on NPRs [4] in a special issue on this topic. Briefly, NPR was identified in a chick oviduct partially purified; its biochemical and physical properties were dependent upon *a prior* estradiol (E_2_) for its cellular synthesis in genetic DNA. Then (1967) P_4_ induced synthesis of a specific target avidin protein in cell culture and in vivo was demonstrated. Steroid hormone stimulated RNA polymerase activity and nuclear expression of ‘new species’ of RNA. E_2_ and P_4_, respectively, could induce the synthesis of the specific mRNAs for ovalbumin and avidin. These discoveries substantiated the overall pathway for intracellular steroid action: steroid hormones could induce the synthesis of specific proteins by inducing the synthesis of their specific mRNAs. When cloning NPR, two NPR isoforms were identified, termed PRα and PRβ, which bind to specific activation sequences for nearby genes. The isoforms are dedicated to differing subsets of P_4_ biology, but cellular levels of both depend upon E_2_. NPR coregulators, coactivators, and corepressors, respectively, enhance and repress transcription. The first nuclear receptor coactivator, steroid receptor coactivator-1 (SRC-1), was discovered to be a type of transcription factor that did not bind to DNA but bound directly to NPRs to activate their transcriptional potential. Nuclear receptors are DNA-binding transcription factors that receive signals and simply locate target genes; NRs recruit coregulators to carry out the enzymatic reactions that are required to activate (or repress). One way to alter NPR action is to alter corepressor exchange with coactivators. An important question in the treatment of many cancers is whether they are steroid receptor positive or negative. Typically, steroid receptor-positive cancers are associated with a better prognosis because there are therapeutic opportunities along the nuclear steroid receptor pathway for intervention and management.

Another key mechanism of P_4_ is through its actions in neurotransmitter systems in the brain. Neurotransmitters, chemical messengers synthesized by neurons, enable connections within and between neurons through ionotropic, metabotropic, electrical, chemical, and paracrine effects. Neurotransmitters traditionally consist of amino acids and their derivatives, chains of amino acids, peptides, or proteins. Steroid hormones also exert paracrine effects on the physiology of neurons and thereby alter the expression of behavior within seconds to minutes. Steroids that bind to neurotransmitter receptors and modulate neurotransmission signaling are termed neurosteroids, those steroids that are both synthesized in the nervous system, either de novo from cholesterol or from steroid hormone precursors, such as P_4,_ that accumulate in the nervous system. Neuroactive steroids are any natural or synthetic steroid that rapidly alters neuronal excitability via nontraditional mechanisms, including P_4_, E_2_, testosterone, glucocorticoid, androstanediol, and allopregnanolone. Other synthetic steroids, such as ganaloxone and brexanolone, possess similar characteristics of endogenous steroids to modulate neuronal activities. These hormonal steroids exert their neuronal excitability functions through various receptors and ion channels, such as membrane or cognate (estrogen) E_2_ receptors, P_4_ receptors, androgen receptors, GABA_A_, DA (Dopamine), NMDA, membrane P_4_ receptors (mPRs), and other receptors [5,6]. See Figure 1.

These neuro(active) steroids are also involved in the etiology, pathophysiology, and/or treatment of various clinical disorders, including but not limited to anxiety [7], depression [8], post-traumatic stress disorder (PTSD) [9], seizure disorders [10,11], various dementias including Alzheimer’s Disease [10], and traumatic brain injuries (TBIs) [11]. Our clinic currently treats these disorders; the predominant clientele is those with TBI, who have post-concussion syndrome, lower levels of neurosteroids in plasma [12], and are treated through behavioral means to manage their symptoms and increase their levels of neurosteroids.

Experimental behavioral approaches typically utilized to understand P_4_’s effects and mechanisms are as follows. First, situations where endogenous changes in P_4_ and its functions are observed for correlations. Second, the effects of extirpation of the ovaries and/or adrenals and the replacement of physiological levels of P_4_ and the effects on behavior/physiology are conducted; this allows a causal relationship to be drawn between P_4_ and its functions. Third, test mechanisms of action by interfering with or mimicking such effects in controlled models. We have conducted many studies of this nature. We reported that P_4_, through its metabolism to allopregnanolone in the hippocampus, improved cognitive performance in the object recognition and T-maze tasks among male, wild-type, and PRKO mice. In the water maze task, c57 wild-type (WT) mice tended to outperform PRKO mice, which were much lighter than the c57 WT controls. Concomitant with these changes were observed increases in brain-derived neurotrophic factor (BDNF) levels in the hippocampus and changes in the half-maximal ^3^[H] muscimol binding and NPR binding with P_4_ in WT, but less consistently, male PRKO mice [13]. To address if there were changes in the effects, along with cognitive performance, we examined animal models of anti-anxiety-like behavior and anti-depressive-like behavior in female WT, PRKO, and 5α-reductase knockout mice (5αRKO) that had been backcrossed to the c57 background strain. The findings shared in this report were run contemporaneously with our 2021 study, which took several years to complete because of the care taken to backcross the strains for 20 generations onto a c57 background for comparability across strains [14]. Study Design is presented in Table 1.

Progestogens are hypothesized to have anti-anxiety and anti-depressive-like effects on behavior, independent of actions at NPRs and dependent on the 5α-reduced, neuroactive metabolite of P_4_, allopregnanolone (3α,5α-THP). What follows are the results of examining estrus cycle differences, effects of extirpation of ovaries, and replacement of E_2_, P_4,_ and/or 3α,5α-THP. The effects on anti-anxiety-like and anti-depressive-like behavior among c57 wildtype (WT), 5αRKO, and PRKO mice and effects on 3α,5α-THP levels in the hippocampus/amygdala complex were assessed. The effects of E_2_ on BDNF levels in the hippocampus/amygdala complex were also examined.

## 2. Result

### 2.1. Mice in Behavioral Estrus Had More Anti-Anxiety-like and Anti-Depressant-like Behavior and Higher Levels of E_2_, P_4_, and Allopregnanolone 

#### 2.1.1. Elevated Plus Maze 

Proestrus mice spent more time in the open arms of the plus maze than diestrus c57 mice. Mice showed significantly more open arm time in the elevated plus maze when in proestrus than diestrus [F (1,19) = 50.31, *p* ≤ 0.001]. Mice in proestrus demonstrated less anxiety-like behavior (i.e., they spent more time on the open arms of the elevated plus maze) compared to control, c57, mice in diestrus (Figure 2, left). Thus, changes in the endogenous hormonal milieu may alter the anxiety-like behavior of c57 female mice.

#### 2.1.2. Forced Swim Test

Proestrus c57 mice spent less time immobile than diestrus c57 mice. Mice in proestrus spent significantly less time immobile in the forced swim test than diestrus mice [F (1,19) = 43.20, *p* ≤ 0.001]. Mice in proestrus demonstrated less depression-like behavior (i.e., they spent less time immobile) compared to control, c57, mice in diestrus (Figure 2, left). Thus, changes in the endogenous hormonal milieu may alter the depression-like behavior of c57 female mice.

#### 2.1.3. Hormone Levels

Proestrus c57 mice had significantly higher levels of steroids in the hippocampus/amygdala complex than diestrus mice. Proestrus mice had significantly higher levels of E_2_ ([F (1,19) = 63.70, *p* ≤ 0.001]), P_4_ ([F (1,19) = 49.24, *p* ≤ 0.001]), and allopregnanolone ([F (1,19) = 52.24, *p* ≤ 0.001]) in the hippocampus/amygdala complex than their diestrus counterparts (Table 2).

### 2.2. Mice with Higher Levels of E_2_, P_4_, and Allopregnanolone in the Hippocampus Showed More Exploratory Behavior in the Open Field 

#### 2.2.1. Open Field–Cycling WT

Proestrus c57 mice made more central entries to the open field than their diestrus counterparts, 5αRKO mice in proestrus, or diestrus c57, WT mice in proestrus. Proestrus c57, WT mice had more anti-anxiety-like behavior in the open field than 5αRKO mice in proestrus and compared to diestrus WT or 5αRKO mice (Figure 3), which was a significant interaction of cycle and strain ([F (1,39) = 34.73, *p* ≤ 0.001]). There were main effects of strain ([F (1,39) = 4.19, *p* ≤ 0.05]) for c57, WT mice to have more central entries than 5αRKO mice. The main effect of the cycle ([F (1,39) = 14.37, *p* ≤ 0.01]) was due to proestrus mice having greater anti-anxiety-like behavior than diestrus mice. 5α-reductase may be necessary for endogenous benefits on E_2_ and P_4_ for affective behavior.

#### 2.2.2. OVX WT or 5αRKO Mice Administered E_2_, P_4_ and or Allopregnanolone & Open Field

Allopregnanolone administration to WT or 5αRKO mice increased central entries to the open field compared to all other groups (Figure 3). Administration of allopregnanolone to WT or 5αRKO mice increased central entries made in the open field compared to all other groups. This was a significant interaction of hormones and strain ([F (3,79) = 22.98, *p* ≤ 0.001]). E_2_ and P_4_ to WT > 5αRKO mice had greater anti-anxiety behavior. OVX E_2_ (0.09 mg/kg) and P_4_-primed 5αRKO mice had lower anti-anxiety-like behavior in the open field compared to control, c57, WT mice. c57 mice made increasingly more central entries compared to 5α-RKO mice at each P_4_ dosage. The main effects of strain ([F (1,79) = 38.91, *p* ≤ 0.001]) were for WT > 5αRKO mice to have increased anti-anxiety-like behavior. The main effects of hormone ([F (3,79) = 35.99, *p* ≤ 0.01]) were due to mice making more central entries as P_4_ dosage increased and for allopregnanolone to have the greatest effects overall. See Figure 4.

#### 2.2.3. Administration of Allopregnanolone to OVX WT or 5αRKO Has Greater Effects on Open Field than P_4_

Administration of allopregnanolone to WT or 5αRKO mice increased central entries made in the open field compared to all other groups. c57 mice made increasingly more central entries compared to 5αRKO mice at each P_4_ dosage; however, allopregnanolone had the greatest effects overall whether administered to WT or 5αRKO mice. This accounted for the significant interaction of hormones and strain ([F (3,79) = 22.86, *p* ≤ 0.001]). E_2_ and P_4_ increased anti-anxiety-like behavior. OVX E_2_ (0.09 mg/kg) and P_4_-primed mice had reduced anti-anxiety-like behavior in the open field. The main effect of hormone ([F (3,79) = 35.99, *p* ≤ 0.001]) was due to mice making more central entries as the P_4_ dosage increased. The main effect of strain ([F (1,79) = 38.91, *p* ≤ 0.001]) was for WT > 5αRKO mice to have more anti-anxiety behavior. Together, these findings suggest actions of 5α-reductase may be necessary for the exogenous benefits of E_2_ and P_4_ for anti-anxiety-like behavior (Figure 5).

#### 2.2.4. Hormone Levels

Adult ovariectomized (OVX), 5αRKO mice had significantly lower hippocampal P_4_ levels when administered P_4_ (4 mg/kg SC) compared to their c57, WT control counterparts. Administration of E_2_ or P_4_ < 3α,5α-THP SC to 5αRKO compared to the vehicle mice to increase 3α,5α-THP in the hippocampus. * = significant decrement in hippocampal 3α,5α-THP levels by E_2_ or P_4_ to 5αRKO vs WT. #5α RKO mice (and P_4_ to WT mice) that were administered 3α,5α-THP SC had higher hippocampal levels of 3α,5α-THP, than all other mice (interaction of strain and hormone; ([F (3,79) = 14.05, *p* ≤ 0.01]). The main effects of hormones were due to 3α,5α-THP producing greater overall effects than E_2_ ([F (3,79) = 13.96, *p* ≤ 0.01]). Strain effects were due to control c57, WT mice having higher hormone levels than 5αRKO mice ([F (1,79) = 75.21, *p* ≤ 0.001]) Table 3.

#### 2.2.5. P_4_ vs Veh Administration to WT and PRKO Mice

PRKO and c57 mice showed less immobility (depression-like) behavior in the forced swim test when administered P_4_ compared to the vehicle.

Administration of P_4_ (4 mg/kg) compared to the vehicle (oil) significantly reduced time spent immobile in the forced swim test ([F (1,39) = 83.60, *p* ≤ 0.001]) (Figure 6). Although there was a main effect of P_4_ administration, there were neither statistically significant differences between the type of mice ([F (1,39) = 2.68, *p* ≤ 0.15]) (WT c57, controls or PRKO KO), nor interactive effects ([F (1,39) = 2.12, *p* ≤ 0.20]) between hormone and mouse type. Thus, P_4_ reduced depression-like behavior, independent of NPRs (Figure 6).

#### 2.2.6. Effects of E_2_ vs Veh Administration on BDNF

c57 mice had greater levels of BDNF when administered E_2_ compared to the vehicles. E_2_ significantly ([F (1,19) = 32.51, *p* ≤ 0.001]) increased levels of BDNF in the hippocampus/amygdala complex compared to that observed in mice administered vehicle (Figure 7).

## 3. Discussion

Our hypothesis that some anti-anxiety and anti-depressant like behavior of mice is independent of NPRs and instead requires the metabolism of P_4_’s neuroactive metabolite, allopregnanolone, was upheld. There were estrus cycle differences in anti-anxiety like behavior in the plus maze and open field and anti-depressant behavior in the forced swim task. c57 mice in behavioral estrus, i.e., proestrus, had higher levels of allopregnanolone in the hippocampus, an important brain area involved in affect and mood, than diestrus mice. Proestrus c57 mice showed more time in the open arms of the elevated plus maze (more anti-anxiety-like behavior) and less time immobile in the forced swim task (less depression behavior). Both c57 and PRKO mice, when OVX and E_2_-primed, responded to P_4_ with similar anti-depressive-like behavior in the forced swim test. There were no differences in the central entries of diestrus c57 and 5αRKO mice. However, proestrus WT c57 and 5αRKO mice differed, with WT c57 mice making more central entries compared to all other groups. The proestrus and 5αRKO mice were intermittent to that of the proestrus and diestrus mice in terms of their open-arm entries, suggesting that 5α-reduction may be necessary for the effects of P_4_. OVX, c57, and 5αRKO mice responded to E_2_ and P_4_ with slightly attenuated anti-anxiety behavior in terms of the frequency of central entries to the open field (c57, WT > 5αRKO). However, when directly comparing the effects of the administration of the vehicle, E_2_, P_4,_ or allopregnanolone to controls (c57, WT) vs 5αRKOs for effects on central entries and hippocampal allopregnanolone levels, allopregnanolone administration increased central entries in the open field of controls and 5αRKOs had increased hippocampal allopregnanolone levels in both groups, but P_4_ did not among 5αRKOs. Estrogen had a modest effect on central entries in both groups and on allopregnanolone levels. The E_2_ regimen had these modest effects in the control but not in 5αRKO mice to increase central entries and elevate allopregnanolone levels, which also increased BDNF levels in the hippocampus compared to the vehicle administration. Together, these findings suggest that some actions of P_4_ for anti-anxiety and anti-depressant-like behavior of experimental mice may be independent of NPRS and involve 5α-reduction, coincident with elevated levels of allopregnanolone in the hippocampus. The actions of E_2_, which primes that of P_4_, can enhance BDNF in the hippocampus. In sum, adult female mice can be responsive to P_4_ for anti-anxiety and anti-depressant behavior, and such effects may be independent of NPRs but require 5α-reduction and actions of BDNF in the hippocampus of E_2_.

These findings confirm the results of other studies and extend the past literature regarding P_4_ and depression-like behavior in preclinical research in some interesting and important ways. First, in the current study, we observed anti-depressant effects in the forced swim task among c57 mice in behavioral estrus with higher levels of E_2_, P_4,_ and allopregnanolone in the hippocampus/amygdala complex compared to that of diestrus c57 mice that demonstrated much more depressive behavior in the forced swim test. Second, OVX PRKO mice administered P_4_ had lower levels of immobility (less depression behavior) and were equivalent to that of OVX c57 mice administered P_4_, and both were significantly lower than vehicle administration to their OVX counterparts. We have previously reported anti-depressant effects of this same P_4_ regimen when administered to c57 or PRKO mice tested in the tail suspension task [15] and that this P_4_ regimen to c57 or PRKO male mice could increase BDNF in the hippocampus [14]. Third, we revealed that levels of allopregnanolone in the hippocampus were elevated in mice administered P_4_ compared to the vehicle. Administration of E_2_ to levels that instantiate allopregnanolone increased BDNF in the hippocampus compared to the vehicle priming. These findings are congruent with the notion that P_4_ has anti-depressant effects independent of NPRs and concomitant with elevated allopregnanolone and BDNF in the hippocampus.

Our findings also confirm and extend the past literature regarding P_4_ and anxiety-like behavior in animal models. We observed anti-anxiety effects in the elevated plus maze with c57 mice in behavioral estrus with higher levels of E_2_ P_4_ and allopregnanolone in the hippocampus compared to that of diestrus c57 mice, which demonstrated much more time on the open arms of the elevated plus maze than their diestrus counterparts. c57 mice in behavioral estrus also showed the greatest number of central entries in an open field, an anti-anxiety effect, compared to proestrus 5αRKO mice > diestrus 5αRKO = proestrus c57 mice. When OVX and E_2_-primed, P_4_ promoted dosage-dependent (4 > 2 = 1 > 0 mg/kg P_4_) increases in central entries in the open field among c57 but not 5αRKO mice. 5αRKO and c57 mice similarly responded to allopregnanolone with maximal increases in central entries and hippocampal allopregnanolone compared to mice administered P_4_ E_2_ or veh. P_4_ (4.0 mg/kg, SC) increased central entries when administered to c57 but not 5αRKO mice; hippocampal allopregnanolone levels followed suit. There were no differences in E_2_ and the vehicle, but both were different from the allopregnanolone administration and the administration of P_4_ to c57s in terms of central entries in the open field and hippocampal allopregnanolone levels. These findings are congruent with the notion that P_4_ has anti-anxiety effects dependent upon 5α-reduction and concomitant with elevated allopregnanolone.

While our findings are interesting and exciting, in light of the need for more new and better anti-depressants, there are reasons to be cautious and temper our enthusiasm. First, our studies were not conducted with induced-knockout mice; they were mice that lacked functional NPRs or 5α-reductase throughout their development, which should have made them aberrant. Given the importance of these knockout models for our work, we made them congenic on a c57 background strain so that their behavior is more comparable to the ‘WT’ c57/UA strain we compared them to. This means that each of the strains has gone through over 20 generations in the UAlbany facility. The latter point is both an asset and a detractor. Between 2006 and 2010, there were fire alarms that were going off at frequent, random intervals in the animal facility. This may have caused a kind of fear conditioning. This information is disclosed in the event there is a trait effect on the mice herein that predisposes them to be more fearful or resilient in response to aversive stimuli. Progestins may have negative effects on the mood among some individuals as a function of their past exposure to steroids, including those resulting from early developmental adverse events or generational adverse events [16]. However, only one mouse was ever used from each litter per condition to minimize cohort or generational effects. Another shortcoming of these studies was that we only measured allopregnanolone in one brain region, the hippocampus/amygdala complex. To clarify, our collection of the hippocampus was a gross dissection of the entire hippocampus, including the amygdala. We collected other tissues (frontal cortex, accumbens, hypothalamus, midbrain, cerebellum, plasma) but did not get to measure these tissues before they were lost in a freezer failure. Further, we do not include PRKOs or 5αRKOs in every experiment for a fully balanced design. We used the -/- PRKOs and -/- 5αRKOs judiciously because only one out of every four bred would be -/-, and only one in eight would be -/- female. As it is, we maximized the information gleaned from these experiments through good design, steadfast work, and patience. We also made a strategic choice to invest in the knockouts to build on studies that have used pharmacological, endocrinological, or acute genetic tools, such as antisense oligonucleotides, to knock down 5α-reductase. The reason is that there is individual variability in response to 5α-reductase inhibitors. About 20% of individuals on finasteride end up having a pathological response, wherein they have a behavioral phenotype that looks like they have had a TBI, where no concussion has occurred [12]. Similarly, about 20% of women on steroids for in vitro therapy adversely react to the hormone regimen [17].

Similar anti-depressant, anti-anxiety, and allopregnanolone-enhancing patterns of effects were observed in our isolated rats in Fairbanks, AK, that demonstrated more exploration and central entries when their diet was supplemented by Native Alaskan blueberries > blueberries from the lower 48 > no blueberry consumption [18]. Our isolated rats eating Native Alaskan blueberries [18], those in the current study administered E_2_, and those in previous studies administered P_4_ [14] demonstrated increases in BDNF in the hippocampus. BDNF alters depression by remodeling in the hippocampus; upstream of that are actions of peroxisome proliferator-activated receptor (PPAR)α at PXR to drive increases in mitochondrial activity and, thereby neurosteroidogenesis to allopregnanolone [19]. There may be similar anti-depressant effects of eating other Native Alaskan delicacies, such as salmon, halibut, and king crab, that are high in beneficial fats due to the cold, arctic waters they come from. Conversely, endocrine disruptors can alter neurosteroidogenesis and, thereby, the cyclicity of women [20]. When women from the Akwesasne Mohawk tribe were followed for daily sampling over multiple months, those with the highest body burden of endocrine disrupters demonstrated reproductive endocrine dysfunction [21]. Similar effects of endocrine disruptors to alter neurosteroid-based endpoints have been observed by a dozen leading laboratories across the world [22]. We and others have also seen adverse effects on anxiety and mood in the adulthood of individuals with maternal separation or adverse events during late pregnancy [23]. Notably, progesterone can obviate the depressant effects of maternal separation and reduce the expression of neuroinflammatory genes (interleukin-1β, IL-1β; tumor necrosis factor α, TNF-α; toll-like receptor 4, TLR4; nucleotide-binding domain, leucine-rich repeat, and pyrin domain-containing protein 3, NLRP3) and decrease the lipid peroxidation (malondialdehyde, MDA) in the hippocampus [23].

Depression is the second leading cause of disability worldwide, with a disproportionate health burden among women [24]. Women are at greater risk during the peri-menstruum, peripuerim, pregnancy, postpartum, and perimenopause, when endogenous hormone levels decline or change. Currently approved pharmacological treatments for major depression include selective noradrenergic inhibitors (SSNRIs), select serotonin inhibitors (SSRIs), and monoamine oxidase inhibitors (MAOIs) [25]. This existing pharmacopeia is problematic. About one-third of individuals will be treatment-resistant to these medications. Another third will struggle with side effects (weight gain, sexual dysfunction, effects on sleep, gut, diet) or problems with access to the medication (difficulty with getting to a physician or pharmacy to get a prescription or cost prohibitions), such that they discontinue the medication before the 4 to 8 weeks latency until their symptoms remit [25]. The other third of patients will improve and perform well. What we and others have observed from those most challenged patients is that those who improve typically have an increase in their allopregnanolone levels along with their depression symptom improvement scores. For example, in a very challenging clinical population, women with the perimenstrual dysphoric disorder (PMDD), their Hamilton Depression Score Ratings (for which a lower score means less depression) were inversely related to their plasma allopregnanolone levels [26,27] when treated with imipramine. In nonclinical populations, there was a correlation between reporting better mood symptoms and consumption of high antioxidants, Native Alaskan blueberries > blueberries from the lower 48 > no blueberry consumption [18]. This was observed among two samples at each location in Fairbanks, Alaska, and Albany, New York [18]. These effects, like those in our present study, are very robust findings.

Variations in E_2_ and/or P_4_ levels across reproductive cycles may increase women’s susceptibility to anxiety and/or depression disorders (mood), which may thereby influence cognition and/or advanced aging. In support, women are uniquely at risk for disorders, such as premenstrual or perimenstrual syndrome [27] (PMS), perinatal or postpartum depression [28] (PPD), and perimenopausal depression [29], which are characterized by their occurrence coincident with robust changes in endogenous hormone levels and/or reproductive status and anxiety and depression symptomatology. Symptoms of PPD typically occur within the first hours, days, or weeks after parturition, when E_2_, P_4_, and allopregnanolone levels precipitously decline from extremely high levels during pregnancy to reach nadir within hours of delivery [28,30]. Extreme steroid-responsive mood disorders, such as PMDD and PPD, often occur in the same women and progressively worsen over time with more steroid exposure [30]. It is not uncommon for a woman with PPD to be restless and unable to sleep if she is anxious, despite just having gone through labor or delivery. This anxiety can be readily heightened if she is unable to manage a seamless transition to bonding and breastfeeding, which can readily turn the tide toward the path of PPD. It is fortunate that women in that situation have an option now: brexanolone, an intravenous synthetic formulation like allopregnanolone. This medication is specifically indicated for PPD only, and it works relatively quickly so that women going into that abyss of a rabbit hole can get out before they reach the point of feeling suicidal or homicidal about their new infant, as occurs when PPD progresses into PMDD. This medication gets women with PPD back on track with the crucial bonding time that they otherwise would never experience with their infant and family [31]. This medication is not without its drawbacks. It is expensive, and there are side effects in some women, such that women must be bed-bound (due to possible ataxic effects) while taking brexanolone infusions for up to 60 hrs. However, women with a history of PMS, PMDD, or previous PPD now have a treatment option.

Brexanolone is a proprietary formulation of synthetic allopregnanolone, a very potent positive allosteric modulator of GABA_A_ receptors [30]. It has long been postulated that the systematic increase in P_4_ and allopregnanolone levels throughout pregnancy has a calming effect on the mother and infant. The sudden decline in allopregnanolone levels following delivery precipitated the onset of PPD. Expression of allopregnanolone-sensitive subunits, α and δ, on GABA_A_ receptors in critical brain regions is significantly reduced during pregnancy. Following delivery, extra-synaptic GABA_A_ receptor δ subunits are involved in maintaining the ratio of excitatory-inhibitory (E/I) balance in the brain kick-in [32]. In murine models, when these δ subunits are removed, aberrant maternal behaviors occur (poor nesting, poor nursing, infanticide), but this effect can be obviated by allopregnanolone administration [33]. An oral synthetic neuroactive steroid, Zuranolone, is being developed for PPD and major depressive disorder (MDD), with encouraging early clinical data. Here, clinical and preclinical findings converged on a pathophysiological target that could be readily altered [34,35]. It is also possible that actions at other targets underlie the effects of Brexanolone, such as membrane mPRs and G-protein-coupled receptors, that are activated by allopregnanolone and linked to the phosphorylation of GABA_A_ subunits. Allopregnanolone enhances the expression of PXR to modulate steroid synthesis and regulate cellular stress mechanisms and BDNF synthesis. Downstream of BDNF, TrkB can interact with anti-depressant drugs through cholesterol [36].

Our findings and the discussion above may suggest that NPRs are not required, but 5α-reduction and allopregnanolone are sufficient for anti-anxiety and/or anti-depressant effects on the behavior of experimental mice. How, then, might one explain the clinical trials of the PR antagonist RU38486 (mifepristone) as an effective treatment of psychotic major depression and other neuropsychiatric disorders [36]? One possibility regards stress and fear-related effects on the hippocampus and amygdala [36,37]. Mifepristone (RU38486) is an effective glucocorticoid receptor (GR) antagonist [34]. In OVX mice given a chronic regimen of E_2_ and/or P_4_, the administration of RU38486 reduced PR expression and anxiety-like behavior [38]. Dysregulation and/or dysfunction of corticosteroid receptors (i.e., mineralocorticoids (MRs) and GRs in the hippocampus and basolateral amygdala [36,37,38,39,40], GRs in the periventricular nucleus, may be involved in the development of depression and anxiety disorders [39]. MRs mediate glucocorticoid action by maintaining homeostasis in the system, whereas GRs are more sensitive to stress-induced increases in glucocorticoids and act to restore disturbances of homeostasis [39]. A dysregulation or dysfunction in GRs and/or MRs may produce the disinhibition of the corticosteroid-releasing hormone (CRH) expression and secretion [40] and thus explain increased HPA function and stress sensitivity in those with depression. Indeed, GR dysregulation occurs in the hippocampus of patients with depression [41,42]. Alternatively, it has been proposed that changes during early development induce altered and persistent HPA feedback that could produce GR resistance in specific HPA feedback regions [43,44] and GR hypersensitivity in other brain regions [44]. Moreover, the amygdala and cortex project to the hippocampus. It has been proposed that such differences may underlie how the impoverished brain gets passed down from generation to generation. Alternatively, RU38486 is a potent ligand for PXR receptors to increase allopregnanolone synthesis, and it could be via this mechanism that RU38486 has its effects [45].

In addition to E_2_ and progestogens altering behavioral plasticity, as described above, some of the effects steroids may have on affective-like behavior may involve neuroplasticity in the hippocampus/amygdala complex. Current clinical studies suggest that neurogenesis in the hippocampus may play a role in depression. Neuroplastic phenomena are involved in the pathophysiology of depression and anti-depressant treatment [46,47]. Hippocampal volume and cell loss are associated with lifetime depressive symptoms through the various aging stages [48]. Therapeutic effects of anti-depressant treatments appear to be neurogenesis-dependent in the dentate gyrus [49]. Animal models of depression also suggest that the administration of anti-depressants increased hippocampal neurogenesis among rats [50,51]. Notably, studies show that allopregnanolone rapidly induced hippocampal neurogenesis [52] through a significant increase in the proliferation of neuroprogenitor cells in the hippocampus of rats and human stem cells [51]. It has been postulated that SSRIs’ anti-depressant effects have less to do with their actions involving serotonin uptake compared to their effects on allopregnanolone or neuroplasticity [51]. These effects to alleviate the symptoms of PMDD, as well as aggression and panic disorders, occur when the concentrations of fluoxetine are in the submicromolar range and have little activity to inhibit the reuptake of serotonin but can alter brain allopregnanolone levels [53]. It may be that therapeutics that have actions to enhance monoamines (e.g., SSRIs, tricyclic anti-depressants) and those that do not (tianeptine; [52]) have their anti-depressant effects by altering the expression of neurotrophic factors, such as BDNF or the formation of new neurons. Unlike the typical timeframe of when the efficacy of anti-depressants is noted among those with depression, SSRIs can have rapid effects to treat relatively short-lasting (but recurring monthly) conditions such as PMDD [53,54,55,56,57]. Thus, these data suggest that the therapeutic potential of steroids’ effects to improve affective behavior may be related to allopregnanolone enhancing neuroplasticity in the hippocampus.

## 4. Conclusions

E_2_, P_4_, and allopregnanolone, like other steroids, have pleiotropic effects across the lifespan. In these experiments, we examined the hypothesis that P_4_’s effects on anxiety and depression behavior in an animal model are through actions of allopregnanolone, do not require actions of NPRs, but can occur with direct SC administration or 5α-reduction from E_2_ or P_4_, which may promote BDNF in the hippocampus. c57 mice in proestrus demonstrated more anti-anxiety like behavior in the elevated plus maze and spent less time in depressive immobility behavior than their diestrus counterparts. Proestrus mice had higher levels of E_2_, P_4_, and 3α,5α-THP in the hippocampus than their diestrus counterparts. Proestrus c57 > 5αRKO mice made more central entries in an open field than diestrus c57 and 5αRKO mice, which were not different. OVX c57 mice administered 1, 2, or 4 mg/kg P_4_ SC showed dosage-dependent increases in central entries to an open field (antianxiety behavior), but 5αRKO mice had maximal increases at 1 to 2 mg/kg P_4_. OVX c57 and 5αRKO mice showed maximum increases in central entries with SC 3α,5α-THP (4 mg/kg), c57’s showed maximal response to P_4_ (4 mg/kg), but 5αRKOs response was half of that. Effects of E_2_ and veh were similar in both groups. Levels of hippocampal allopregnanolone coincided with effects on central entries in the open field. P_4_ (4.0 mg/kg SC) to OVX c57 or PRKO mice had decreased immobility (depression behavior) in the forced swim task. Levels of hippocampal BDNF were greater among E_2_-primed c57 mice compared to the vehicle-administered mice. In sum, adult female mice can be responsive to P_4_ for anti-anxiety and anti-depressant behavior, and such effects may be independent of NPRs but require the 5α-reduction and actions of BDNF in the hippocampus of E_2_. Allopregnanolone is now the target molecule of a new slate of therapeutics and can be used as a pharmacological and behavioral tool to help individuals with neurodevelopment, neuropsychiatric, and/or neurodegenerative sequelae.

## 5. Materials and Methods

The methods and protocols utilized for animal husbandry, genotyping (for the determination of homozygous PRKOs, 5αRKOs), drug administration, behavioral testing, euthanasia, and tissue collection in the murine subjects in this study were as previously described in reports from our lab in this facility and were approved by the Institutional Animal Care and Use Committee at the University at Albany. The experiments herein were approved in protocol #08-17.

Colony Maintenance: Using heterozygous breeder pairs in a continuous mating situation, c57, PRKO, and 5αRKO littermates were successfully bred and maintained in the Animal Facility at The University of Albany as previously described. Litter/cohort effects were minimized because mice came from many litters across time.

Animal Husbandry: Subjects were adults (12 to 20 weeks old) and female mice. Mice were group-housed (4 to 5 per cage) in polycarbonate cages (26 × 16 × 12 cm) in a temperature-controlled room (21 ± 1 °C) in the core Laboratory Animal Care Facility at The University at Albany. The housing room was on a 12/12-h reversed light cycle (lights off 8:00 a.m. to 8:00 p.m.). Mice had continuous access to Purina Mouse Chow and tap water in their home cages and were assessed during their dark phase.

Mouse Strains and Genotyping: Female wildtype (WT; N = 180) or homozygous (*n* = 120) 5α-reductase knockout (5αRKO) or PRKO mice (*n* = 20) were derived from heterozygous breeder pairs. 5αRKO mice, originally characterized by Mahendroo’s lab, are genetically deficient in the 5α-reductase enzyme, and they have low levels of allopregnanolone in the brain during proestrus and after P_4_ administration. PRKO heterozygous breeder pairs were generously provided by Dr. Bert O’Malley. Genomic DNA was isolated from tails and analyzed by PCR in our laboratory and/or the Molecular Core Facility at The University of Albany per previously described methods [15,54].

Summary of Animal Use: There were N = 180 c57UA, N = 120 5αRKO, N = 20 PRKO mice, *n* = 10 mice/group used in all of these studies. In experiments 1.1.1 and 1.1.2, and 1.1.3, 20 cycling c57UA mice were used. In experiment 2.2.1, there were 20 c57UA and 20 5αRKO mice used. In experiments 2.2.2, 2.2.3, and 2.2.4, there were 100 c57UA and 100 5αRKO OVX mice used. In experiment 2.2.5, 20 c57UA and 20 PRKO mice were used. In experiment 2.2.6. 20 c57UA mice were used. In sum, a total of 180 c57UA mice were utilized, 120 homozygous 5αRKOs and 20 homozygous PRKOs. A summary of experiments is shown in Table 1.

General health and normative response screening procedure: To verify that all experimental mice show normative responses before initiation in behavioral studies, adult female mice, at least 60 and no more than 120 days of age, were evaluated for general health and normative responses to stimuli, using modified methods [58,59]. To determine the health status of mice, daily observations of their general appearance, normative behavior, and normative sensory responses were conducted. Only mice that did not demonstrate deviations from norms for these measures were included in the study.

Habituation Procedure: The behavior of mice is very sensitive to arousal [58,59,60,61,62]. To minimize the potential effects of arousal and habituate mice to steroid injections, handling, and behavioral testing, mice were habituated in the following manner for the first week [60]. Briefly, mice were picked up from their homecage, handled for 15 secs, and returned to their homecage (Day 1). Mice were transferred from their homecage to a novel cage (Day 2) and weighed and then returned to their homecage (Day 3), transferred to another room via a cart (Day 4), and transferred to another room via a cart, injected with 0.2 cc vegetable oil SC, and placed in novel environment for 5 min (Day 5).

Manipulations: All manipulations have been successfully utilized in adult female mice in our laboratory.

Cycling: Some mice had vaginal opening or cytology, analyzed to determine behavioral estrus, i.e., proestrus).

OVX: Some mice were OVX under anesthesia (sodium pentobarbital; 70 mg/kg; IP [58,59,60,61,62] and controls were sham OVX, which means they received anesthesia and an incision in the skin and abdominal wall. Mice did not begin behavioral testing until at least 10 days had passed.

Systemic steroid administration: E_2_ (0.09 mg/kg, SC) at time 0 and SC P_4_ (1.0, 2.0 or 4.0 mg/kg, SC), 3α,5α-THP (4.0 mg/kg, SC) or vehicle (vegetable oil or β-cyclodextrin) were injected SC in the back of the neck 42 h later from E_2_, and testing occurred between 3 and 6 h after progestogens. These injection and timing regimens reproduce levels that are akin to those observed during behavioral estrus (proestrus) of mice. Steroids were obtained from Steraloids, Inc., Newport, RI, USA.

Data Collection: Observers collected behavioral data using a computer-assisted video-tracking program (Any-Maze, Stoelting Co., Wood Dale, IL, USA). Observers were uninformed of experimental conditions by coding hormones and genotype. During behavioral testing, mice were tested in the early phase of the dark cycle (1 to 3 hr after lights off) once each week for up to four weeks. The first week, they were habituated. Then, they were tested once a week for up to three weeks, and in the final week, tissues were collected. Mice were tested in the elevated plus maze, in the forced swim test, and/or in the open field task. The order in which mice were tested in each set of tasks was counterbalanced. In all tasks, control/motor measures are readily obtained (i.e., total square or arm entries in the open field, plus maze, respectively, struggling versus immobility in forced swim test), which allows for the dissociation of effects of E_2_/progestogens on mood vs motor behavior. Motor behavior did not account for any of the results observed.

Elevated Plus Maze: The matte black stainless steel elevated plus maze (Columbus Instruments, Inc., Columbus, OH, USA) has two open arms, which are 30 cm in length and 5 cm in width, and two closed arms, which are the same size, but enclosed by 14.5 cm high walls. The arms of the maze are elevated 47.5 cm off the ground. Briefly, mice were placed at the intersection of the open and closed arms, and the time spent and the number of entries to the open and closed arms were recorded during a two minute test [59,60,61]. Increases in open arm time were considered a reflection of anti-anxiety-like behavior.

Forced Swim Test: Mice were placed in a glass cylinder obtained from Stoelting, Inc. (Wood Dale, IL, USA), which was 20.5 cm in diameter and 21.5 cm in depth, filled with 18 cm of 25 °C water for 300 secs. Time spent immobile was recorded. There were 40 c57, WT, controls, and 40 5αRKO that were also in experiments. The forced swim test is an often-utilized assay of depressive behavior in rodents [63]. Behavior in these tasks is sensitive to the effects of ovarian hormones, and our laboratory has used this task successfully to investigate these effects. Behavior in the forced swim task is mediated by the hippocampal complex. In both, a reduction in time spent immobile and increased time spent struggling or swimming reflected antidepressive behavior, with strong reliability and predictive validity. Immobility was readily identifiable and quantifiable as the absence of active behaviors (i.e., swimming, jumping, or diving) following placement in the water testing chamber.

Open Field: The motor behavior of mice was assessed in a 39 × 39 × 30 cm activity monitor (AccuScan Instruments, Inc., Columbus, OH, USA), as per previous methods [61,62,64]. The activity chamber had a grid floor with a total of 16 equal squares delineated. An observer recorded the number of entries made by the mouse into the 12 peripheral or four central squares for 5 min, whilst interruptions in light beams in a horizontal plane were automatically recorded. Increased central entries are indicative of anti-anxiety-like behavior. Total square entries are indicative of general motor behavior.

Tissue Collection: Tails were ‘snipped’ for genotyping. Blood obtained from the trunk and/or cardiac puncture and brains were obtained and used for endocrine and BDNF analyses, as described below.

RIA: Using previously described methods, E_2_, P_4_, and Allopregnanolone were measured by RIA [60,61] in the plasma, hippocampus, and cortex to verify conditions and elucidate the relationship between performance and steroid levels. Ether was added to plasma to extract steroids. E_2_ and progestogens were extracted from brain tissue with methanol and chromatographed on Sepak cartridges. Tritiated probes (Perkin-Elmer): E_2_ (NET-317, 51.3 Ci/mmol), P_4_ (NET-208, 47.5 Ci/mmol), and Allopregnanolone (for Allopregnanolone, NET-1047, 65.0 Ci/mmol) were added to samples. Standard curves were prepared in duplicate. Antibodies: 1:20,000 dilution, E_2_ (E#244, Dr. Niswender, Colorado State Univ.) in a 1:40,000 dilution, P_4_ (#337, Dr. Niswender) in a 1:30,000 dilution, and Allopregnanolone (921412-5; Dr. Robert Purdy, Scrips, La Jola, CA, USA) 1:5000 dilution were added. After incubation, the bond was separated with dextran-coated charcoal. Sample tube concentrations were calculated using the logit-log interpolation of the standards and correction for recovery. The intra- and inter-assay coefficients of variance were: E_2_ 0.07, 0.08; P_4_ 0.08, 0.09; Allopregnanolone 0.11, 0.13.

BDNF measurement: The kit was a BDNF Emax Immunoassay system (G7610) designed for the sensitive and specific detection of BDNF in a sandwich format. Specific detection of BDNF is typically less than 3% less cross-reactivity of other neurotrophic factors at 100 ng/mL. Sensitivity detects a minimum of 15.6 pg.ml. The hippocampus/amygdala was dissected bilaterally from all mice and analyzed for BDNF. Briefly, tissues were homogenized in distilled water and cell lysis buffer (Qiagen; Germantown, MD, USA). Fifty microliters of homogenates were diluted in four volumes of Dulbecco’s Phosphate-Buffered Saline. Diluted samples were acid treated by adding 1 microliter of 1N HCl, incubating for 15 min, and then neutralizing the samples by adding 1 microliter of 1N NaOH. Analyses of BDNF were performed using standard methods of the Emax Immunoassay system (Promega; Madison, WI, USA), as previously utilized [14]. Protein was measured using a Nanodrop Spectrometer with absorbance at 450 nM.

Statistical analyses: For all measures, one-way or two-way analyses of variances were used to examine the effects of hormone condition and/or genotype on behavioral, hormonal, or BDNF endpoints. When the α level for statistical significance was *p* ≤ 0.05, Tukey’s *post hoc* tests were used to determine groups that were different.

## Figures and Tables

**Figure 1 ijms-26-01173-f001:**
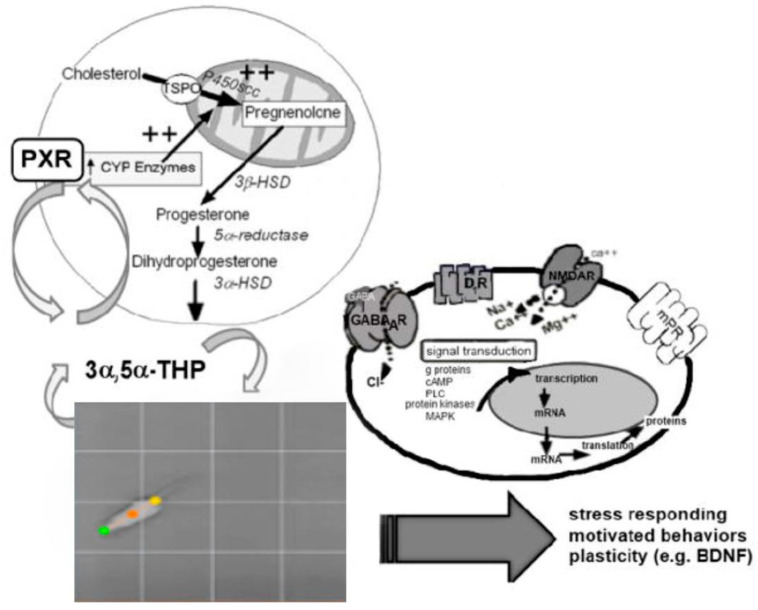
Putative pathways of allopregnanolone (3α,5α-THP) sources (left side) and some targets of action (right side). 3α,5α-THP can be metabolized in the brain from progesterone by 3α (β)-hydroxysteroid dehydrogenase and 5α-reductase, which estradiol can upregulate. Allopregnanolone is secreted de novo in response to challenges (e.g., open field behavior) that activate the autonomic nervous system to reinstate parasympathetic tone, in part by upregulating pregnane xenobiotic receptor function (PXR) in brain. Targets of allopregnanolone’s actions include steroid binding sites at GABA_A_, dopamine, NMDA, and cognate progestin membrane receptors (mPRs) (others not shown), but not nuclear progesterone receptors (NPRs) in physiological concentrations [5,6].

**Figure 2 ijms-26-01173-f002:**
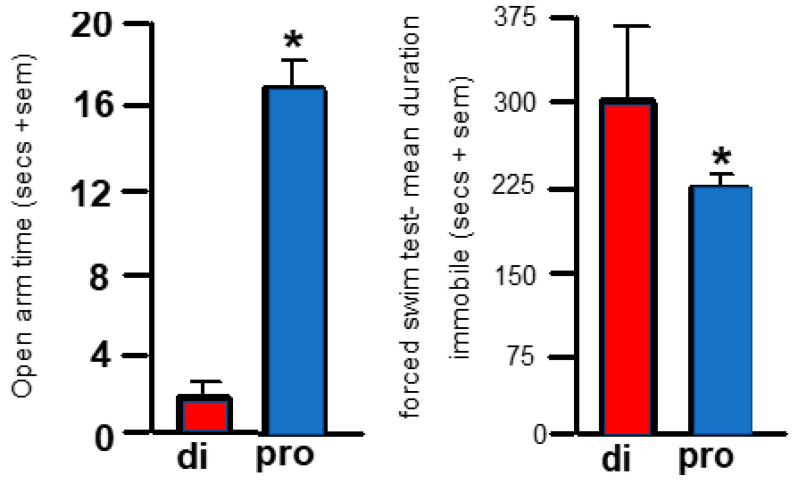
The (**left panel**) shows anti-anxiety-like behavior in the elevated plus maze. * Time spent in open arms is significantly greater in proestrus (pro; c57Bl/6J (c57) mice indicated in blue bars (*n* = 10) than diestrus (di, c57 mice indicated in red bars, *n* = 10) in a 120 sec test. The (**right panel**) shows anti-depressant-like behavior in the forced swim test of 300 secs. c57 mice in proestrus (*n* = 10) spent less time immobile than diestrus mice (*n* = 10). Immobility is considered a sign that one has given up; it is a proxy measure of depression. * represents a difference between groups is greater than *p* ≤ 0.001.

**Figure 3 ijms-26-01173-f003:**
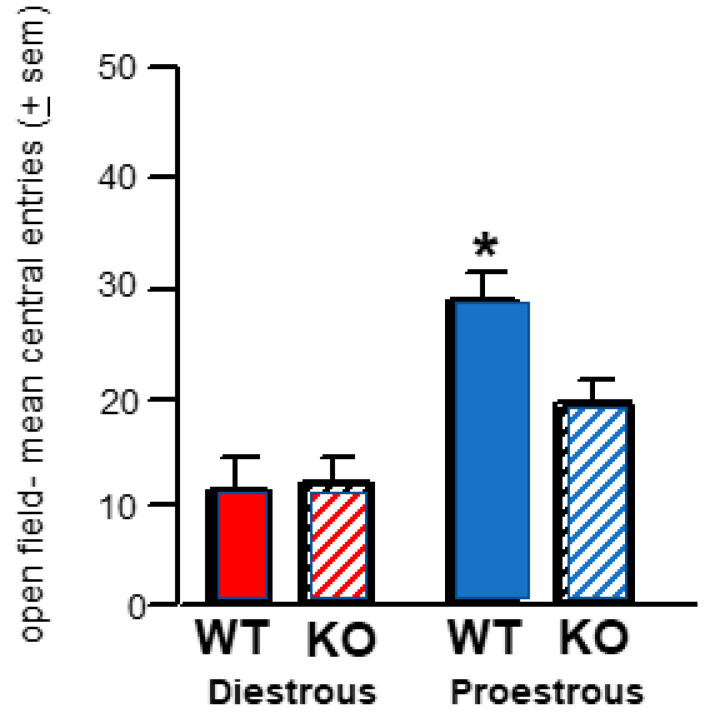
Diestrus (left side; red) and proestrus (right side; blue) control, c57, wildtype (WT) mice are depicted by solid bars, and 5αRKO mice are depicted by bars with horizontal stripes; *n* = 10/group. * There was a significant *p* ≤ 0.001 interaction of cycle and strain as c57, WT mice in proestrus made more central entries compared to all other groups. There were main effects of strain for WT, c57 mice to have more central entries than 5αRKO knockout mice and main effects of the cycle for proestrus mice to have more central entries than diestrus mice.

**Figure 4 ijms-26-01173-f004:**
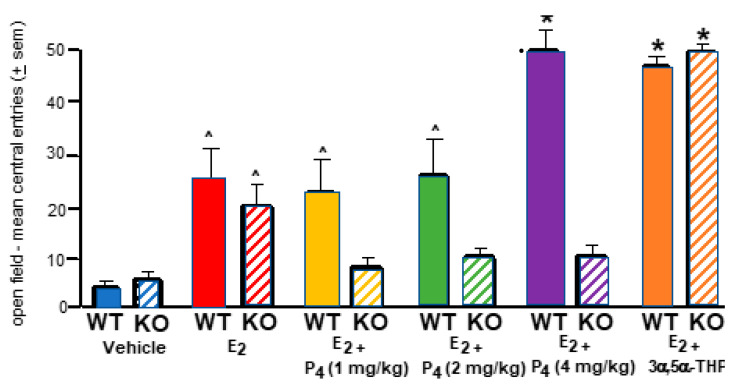
WT (colored bars) and 5αRKO deficient mice (striped bars) were OVX and administered oil vehicle or E_2_ (0.09 mg/kg) and different dosages of P_4_ (1 (yellow), 2 (green), or 4 mg/kg (purple) or 3α,5α-THP (4 mg/kg; orange) SC. * Administration of 3α,5α-THP to WT (c57) or 5αRKO mice increased central entries to the open field comparable to that of WT mice administered E_2_ and P_4_ (solid purple) and greater than that of all other groups. There was the significant interaction of hormones and strain. E_2_ (red), compared to the vehicle (blue), increased central entries of WT and 5αRKO mice, E_2_ and P_4_ (1 (yellow) or 2 mg/kg (green) increased central entries of WT mice, a main effect of hormones, ^ represented by the upside-down caret.

**Figure 5 ijms-26-01173-f005:**
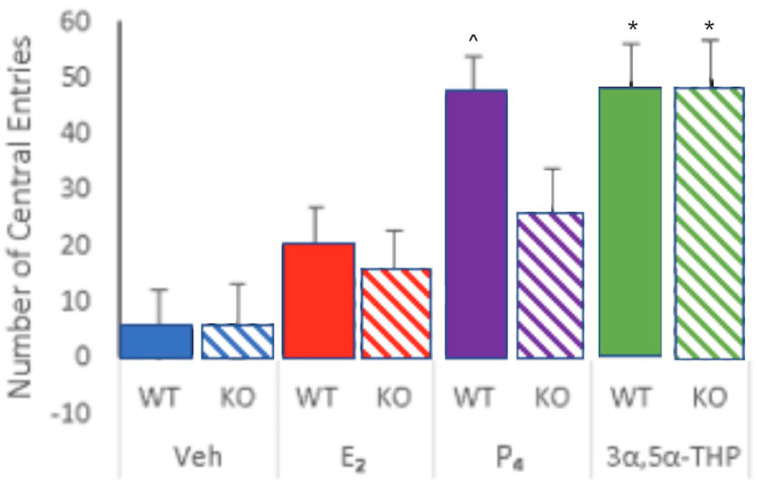
c57, wildtype (WT mice are depicted by solid-colored bars, and 5αRKO mice are depicted by colored bars with horizontal stripes; *n* = 10/group. * There is a significant interaction of hormone and strain as WT and 5αRKO mice had robust responses to allopregnanolone on central entries to the open field compared to P_4_, to WT and 5αRKO mice. There were main effects of the hormone condition for E_2_, P_4_, and 3α,5α-THP (allopregnanolone) to produce significant increases compared to the vehicle irrespective of. The ^ indicates the significant differences between WT and 5αRKO at this P_4_ dosage. The main effect of strain was for WT to have more effects than that of 5αRKO mice on central entries.

**Figure 6 ijms-26-01173-f006:**
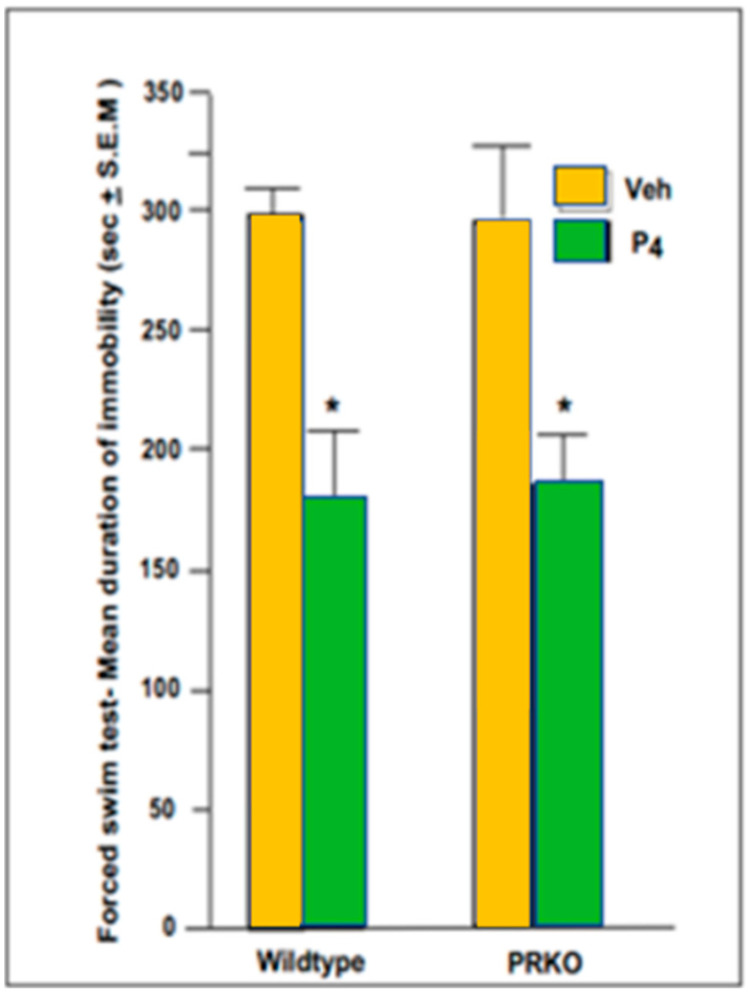
c57/Wildtype (left side) and PRKO (right side) mice showed less immobility and depression-like behavior in the forced swim test when administered 4 mg/kg of progesterone; P_4_ (green bars) compared to when administered oil vehicle (yellow bars). There are *n* = 10/group. * There was a significant main effect of P_4_ to decrease immobility, but neither an effect of mouse strain nor an interaction of what was administered and strain.

**Figure 7 ijms-26-01173-f007:**
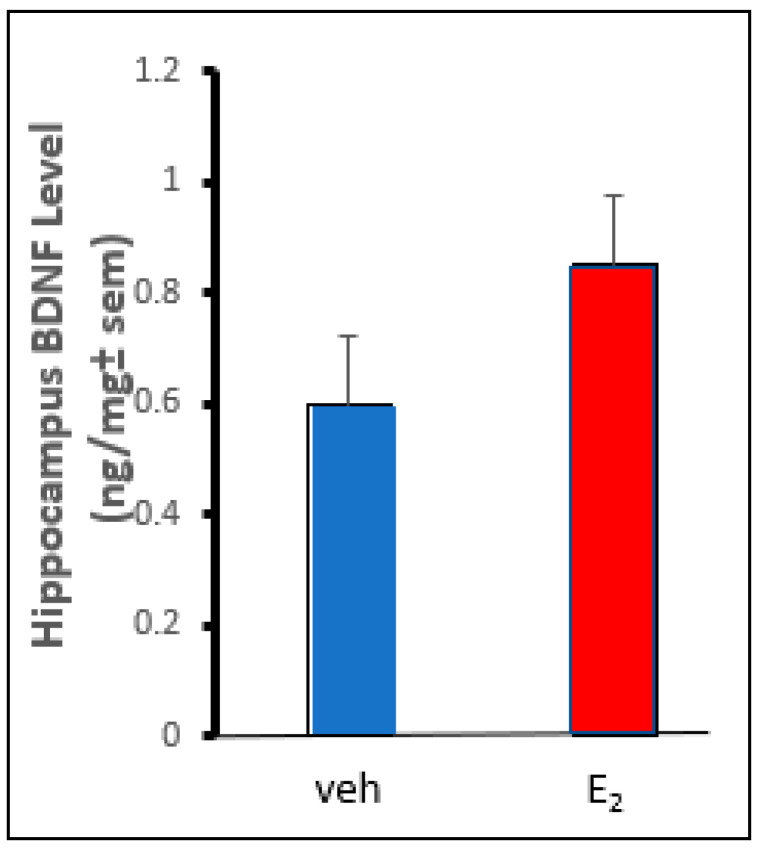
Hippocampus/amygdala-complex brain-derived neurotrophic factor (BDNF) levels in the hippocampus of c57 mice administered vehicle (blue bars) vs mice administered (red bar) E_2_ SC. (*n* = 10/group). There was a main effect of estradiol (E_2;_ red bar on the right) to increased BDNF in the hippocampus/amygdala-complex compared to the vehicle (veh; blue bar on the left, *n* = 10 per group).

**Table 1 ijms-26-01173-t001:** Summary of experiments.

Study Number #	Number and Type of Mice Used	Estrus Cycle or OVX Hormone-Primed Condition to Mimic Cycle	Behavior, Hormone, or Trophic Factor Measures
Study #	Number and type of mice	Independent Measures	Dependent Measures
1.1.1	*n* = 20 c57UA mice	Diestrus vs proestrus	Elevated plus maze
1.1.2	Same as above	Diestrus vs proestrus	Forced swim test
1.1.3	Same as above	Diestrus vs proestrus	Hippocampal/amygdala hormones
2.2.1	*n* = 20 c57UA mice, *n* = 20 5α-reductase knockout mice	Diestrus vs proestrus	Central entries to Open field
2.2.2	*n* = 60 OVX c57UA mice; *n* = 60 OVX 5α-reductase knockout mice	E_2_ (0.09 mg/kg SC) and P_4_ (0, 1, 2, 4 mg/kg) or 3α,5α-THP (4 mg/kg)	Central entries to Open field
2.2.3	*n* = 40 OVX c57UA mice; *n* = 40 OVX 5α -reductase knockout mice	Veh vs E_2_ (0.09 mg/kg SC) and P_4_ (4 mg/kg) or 3α,5α-THP (4 mg/kg)	Central entries to Open field
2.2.4	Same as above	Veh vs E_2_ (0.09 mg/kg SC) and P_4_ (4 mg/kg) or 3α,5α-THP (4 mg/kg)	Hippocampal/amygdala hormones
2.2.5	20 PRKO mice, 20 c57UA mice	Veh vs P_4_ to PRKO or c57UA mice	Forced swim test
2.2.6	20 c57UA mice	E_2_ (0.09 mg/kg SC) or veh	Hippocampal/amygdala BDNF

**Table 2 ijms-26-01173-t002:** Levels per mass of protein of estradiol, P_4_, and its 5α-reduced metabolite, allopregnanolone, are significantly higher (* = *p* ≤ 0.001) in the hippocampus/amygdala complex of proestrus (*n* = 10) than diestrus (*n* = 10) c57 mice.

Cycle Phase	Estradiol (pg/g)	P_4_ (ng/g)	3α,5α-THP (ng/g)
diestrus	6.4 ± 1.3	0.8 ± 0.3	3.8 ± 0.3
proestrus	14.7 ± 1.3 *	16.4 4.3 *	20.4 ± 2.4 *

**Table 3 ijms-26-01173-t003:** 3α,5α-THP levels in hippocampus/amygdala complex of OVX mice administered vehicle (oil or β-cyclodextrin; each dosage was administered in 0.1 cc, SC), E_2_ (0.09 mg/kg, SC, ~48 h before tissue collection) P_4_ (4 mg/kg, SC, 3 to 6 h prior to tissue collection) or 3α,5α-THP (4 mg/kg, SC, 3 to 6 h prior to tissue collection). * *p* ≤ 0.01 different from c57, WT control at this hormone condition. # means significantly different than vehicle condition.

Mice Levels of 3α,5α-THP (ng/g) of Protein in Hippocampus/Amygdala Complex							
Veh		E_2_		P_4_		3α,5α-THP	
WT	5αRKO	WT	5αRKO	WT	5αRKO	WT	5αRKO
1.5 ± 0.4	1.4 ± 0.6	3.9 ± 1.0	* 2.1 ± 1.1	#7.6 ± 1.	* 4.9 ± 2.2	#8.9 ±1.6	#8.6 ± 1.1

## Data Availability

Data are publicly unavailable due to privacy and ethical restrictions.

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
