# Peer review of "Progesterone Promotes Anti-Anxiety/Depressant-like Behavior and Trophic Actions of BDNF in the Hippocampus of Female Nuclear Progesterone Receptor, but Not 5α-Reductase, Knockout Mice"

_ijms, 2025, doi:10.3390/ijms26031173_

Round 1
Reviewer 1 Report (New Reviewer)
Comments and Suggestions for Authors
Manuscript titled “Progesterone promotes anti-anxiety/depressant behavior and trophic actions of BDNF in the hippocampus of Female Nuclear Progesterone Receptor, but not 5α-reductase, Knockout mice” reports the effects of progestogens on KO mice, according to behavioral anxiety and depression models. There are comments and suggestions for the authors:
1. Lines 38-48 begin with “among males”, and end with the same words, thus, please consider eliminating one to avoid repetition.
2. Lines 49-66 mention the contributions of Dr. O’Malley to the field. It appears that the authors have a very high opinion of him and his contributions, however, please consider writing this section in a more neutral tone, since it currently reads like a personal opinion or in a tone that does not appear to be suitable for the introduction of a scientific paper. This also distracts from the main scientific message that the authors are trying to convey to the reader. Dr. O’Malley is thanked in acknowledgements; the authors’ could express further gratitude in this section if they so desire.
3. The aforementioned section mentions some specific contributions, but the references are not properly given, for example, results are mentioned from 1967 and 1972, but the actual reference is not found in the reference section.
4. Line 65 mentions two forms of PR, but their subscripts are apparently corrupted, since only a spiral is shown. Please check these subscripts. Also check throughout the document, since this error appears multiple times.
5. Line 94 references a diagram (apparently the one in line 367). Please rename this to “Figure” instead of “Diagram”, number it accordingly, provide its caption on the figure legend (not on the image itself), and move it to where it is initially referenced instead of at the end of the document.
6. In tables 1 and 2, units are shown “per gram”, however, it is not clear that this gram is (protein, tissue, etc.). Please clarify. Same question for figure 6, where “ng/mg” are shown.
7. Figure 4 contains no indications of statistical significance on any bar, although it appears that there is a significant difference on P4; please confirm if this is the case.
8. Please homogenize the font, color and overall style of all graphs, in order to look consistent between them.
9. There is an unnumbered table in page 14, which apparently lists the number of animals used on each experiment. Please reference the table in the main text, number it accordingly and provide its title and footer.
10. Line 554 mentions “aka proestrus, when a noun, or proestrous, when an adjective”. Is this information necessary, or could it be omitted?
11. Please mention the manufacturer for the AnyMaze software (line 565) and the “apparatus” (line 566).
12. Please specify the manufacturer for E2, P4 and 3alpha 5alpha THP (lines 559-560).
Author Response
Please see the attachment.

Reviewer 2 Report (New Reviewer)
Comments and Suggestions for Authors
· Paper by Frye is well planned, executed, presented, and formatted. Some comments:
· Title: because this animal study I suggest to add like anti-anxiety/depressant like behavior
· Abstract: delete (e.g., progesterone; P4 and its metabolites, allopregnanolone)
· Lines 49-53 need to be phrased for academic reading e.g. terms like Distinguished Professor and generously makes PRKOs available can be deleted
· Line 107: Missing ref also expand how allopregnanolone modulates neural circuits or neurotransmitter systems (for example GABAergic pathways) to induce behavioral changes
· Line 112: Spell BDNF to full
· Line 112: The role of BDNF in mediating the behavioral effects is not entirely clear if E2 enhances hippocampal BDNF levels how will this be anxiolytic/antidepressant
· Line 121: Check symbols of alphas
· Line 135: (As example) Statistical analyses are comprehensive but some instances lack clarity e.g. P <=0.01 pls report exact p this important for future reviews/meta analyses. P also need to be p
· Figures can be improved and perhaps colored
· While most abbreviations are defined, some (e.g., E2, BDNF) are introduced without explanation. Ensure all abbreviations are defined upon first use
· Discussion to apply these findings what can be offered for clinical populations, such as individuals with anxiety, depression, or traumatic brain injury see https://www.nature.com/articles/s41386-023-01626-z
Some refs to be added + improve referencing system
· https://pmc.ncbi.nlm.nih.gov/articles/PMC6968520/
· https://www.mdpi.com/1424-8247/16/4/520
Improve ref system = e.g. Line 107: Missing ref the authors self-cite their work without mentioning explicitly in which paper is reported.
Author Response
please see the attachment

Reviewer 3 Report (New Reviewer)
Comments and Suggestions for Authors
- Innovative Focus: The study addresses an important gap by elucidating the mechanisms underlying P4's behavioral effects, advancing neuroendocrine research.
- Comprehensive Design: The use of multiple mouse models (WT, PRKO, 5αRKO) strengthens the mechanistic insights.
- Clear Hypothesis Testing: The experiments align well with the stated hypothesis, providing compelling evidence for the role of 5α-reduction.
Critiques:
- Data Presentation:
- Figures lack clarity in labels and legends. For example, Figures 2 and 3 require more precise annotations to highlight significant trends and group differences.
- Consider improving figure captions by clearly describing the experimental groups and statistical significance markers.
- Data points are necessary for all figures (1 to 6) to enable verification and replication. Without these, the visual presentation is incomplete.
- The last figure, showing the putative pathway, is unclear and inadequately labeled. It is difficult to interpret and requires expansion with proper annotations. This figure should also be relabeled as Figure 7.
- Methodological Limitations:
- The analysis of allopregnanolone was restricted to the hippocampus, neglecting other regions critical to mood regulation (e.g., prefrontal cortex, amygdala).
- The imbalance in sample sizes (e.g., only 20 PRKO mice versus 140 WT mice) raises concerns about statistical robustness.
- Discussion and Interpretation:
- The environmental stressors (e.g., fire alarms) affecting animal behavior are acknowledged but not sufficiently controlled or accounted for in the analysis.
- The extrapolation of findings to human conditions like PMDD and PPD is speculative without stronger translational evidence.
Suggestions:
- Expand tissue sampling to include additional brain regions to capture a more holistic view of P4’s neurophysiological effects.
- Provide more detailed statistical methods, including how randomization and blinding were ensured during data collection.
- Include a clearer discussion of the limitations of using genetically modified models for long-term behavioral studies.
- Revise the putative pathway figure (Figure 7) to make it more comprehensive, with clear and appropriately labeled components.
Round 2
Reviewer 3 Report (New Reviewer)
Comments and Suggestions for Authors
NA
Author Response
1) fix the alpha in last line of conclusions
- Thank you for pointing out this error; line 544 has been edited.
2) the conformation in the formatting of the refs is off, such that some have all words in title in caps and others don't
- I appreciate your thorough review, and the suggested capitalization corrections have been made in line 716.
- Thank you for pointing out this formatting mistake; lines 724 and 725 have been edited.
- The reference in lines 735-736 was corrected.
- Your suggestion improved clarity; lines 735-737 have been edited.
- Your feedback helped refine this section; lines 739-740 have been edited.
- I’ve made edits to lines 742-743 based on your helpful feedback.
- The reference in line 743-744 has been amended.
- The clarity of lines 750-751 have been improved according to your suggestions.
- Your feedback has made lines 756-758 more clear.
- Your note was helpful; lines 791-792 have been revised.
3) we are inconsistent in use of 5a-RKO vs 5aRKO no dash in the statistics. With the a here being alphas
- Consistency is important to overall clarity and I appreciate you noting this; line 189 has been edited to maintain no dash consistency.
- Thank you for making note of this inconsistency; line 207 has been edited.
- Thank you pointing out this issue; line 213 has been edited.
5) there are a few P4 I saw that still need to be subscripted.
- Thank you for pointing out these inconsistencies; the subscripts in lines 656 and 669 have been edited.
This manuscript is a resubmission of an earlier submission. The following is a list of the peer review reports and author responses from that submission.
Round 1
Reviewer 1 Report
Comments and Suggestions for Authors
Review for Int J Molec Sci
Progesterone promotes anti-anxiety/depressant behavior and trophic actions of BDNF in the hippocampus of Nuclear Progesterone Receptor, but not 5a-reductase, Knockout mice
Dr. Cheryl Anne Frye and Colleagues
01/15/2024
Progesterone is an essential hormone that has numerous effects on both men and women across the lifespan. Understanding the role of hormone mediated behavior and disease is important and not well studied. Frye and colleagues address this gap in the literature using mouse models of progesterone and a series of validated mouse behavioral paradigms (e.g. forced swim test, open-field test, elevated plus maze). Investigations into hormones and behavior, and also hormones and drug interactions, are, I believe, important and long overdue in animal and human research.
The introduction includes a historical review of O’Malley’s investigation into the progesterone receptor (PR) and the basis for testing the effects of progesterone in wild-type, PR deficient, and OVX 5a-R deficient mice. Frye and colleagues then provide interesting data that seem to support increased levels of 3a,5a-THP (ng/g), Estradiol, and Progesterone steroids in the hippocampus of female mice in proestrus vs diestrus phases. The hypothesis that fluctuations in estrous cycle that align with high levels of progesterone are also associated with lower anti-anxiety/depressant behavior in female mice independent of nuclear PRs was also supported using PRKO mice. Finally, ovariectomized wild-type mice, but not OVX 5a-R deficient mice, injected with estradiol or progesterone replacement showed less anxiety (increased center entries, decreased immobility), suggesting a role for 5aR in processing progesterone-mediated affective behavior. Overall, the authors suggest a role for progesterone and estradiol in anxiety and depression behaviors that are associated with increased levels of BDNF and allopregnanolone in the hippocampus, with important implications for mood and health.
However, the paper is not suitable for publication in its current form.
Major Concerns:
1. Introduction/Discussion: The authors show a deep understanding of past literature and present their hypotheses clearly; unsurprisingly, they have several other papers on progesterone’s effects. However, it has no review of current literature, with most papers from the 1990s or 2000s (other than the O Malley 2020 paper). The manuscript reads more like an old grant proposal and is need of updating.
· The paper requires a major overhaul of the literature discussed to include the current relevant literature. For example, line 217 refers ‘recent clinical trials’ and cites a 2006 paper
· The first paragraph of the discussion does not summarize the results of the current experiments in enough detail.
· The discussion is too long and would benefit from the authors reframing the discussion to more directly reflect how the findings of the current paper fit back into the recent literature. For example, I am very concerned that the manuscript is quite similar to the Frye, Lembo and Walf, 2020 paper, and yet no interpretation of current results is made in the context of the author’s own recent work
Methods and Results:
The current paper contains many details, including the validation of the employed animal paradigms. However, the manuscript seems to have unnecessary details and seems to be missing necessary details. With what is currently provided, there is no way to assess scientific soundness. The following info is needed:
· More details on subjects. I cannot find the subject ages, number in each group, their husbandry (e.g. group density, environment, and general care, IACUC). Maybe there are supplemental materials I am missing?
· I also do not see any statistical methods explaining how significance was determined or any results that contain statistical descriptions with degrees of freedom, p-values, etc. Since no raw data are provided, this is a major concern.
· the manuscript would benefit from editing for additional citations where needed (e.g. line 39, line 192, line 226)
· There is often a lack of agreement in tense, between past and future. Please edit and put all methods in past tense. For example, line 489 ‘Plates will be blocked… and line 490 ‘Plates were washed…’
· The figure captions seem to be incomplete (e.g. Fig 2) and the information that is in each caption is not provided in enough detail making them difficult to interpret, e.g. fig 1, what are the colors indicating? To all other figures, please add subject numbers
· The authors say why using mice and rats is a strength of their research program (lines 364-368), which seems unnecessary for this paper.
· Cycling and OVX sections do not have enough detail. When did you sample estrus? Before or after behavioral tests? How long did mice have to recover from OVX? How many mice were OVX vs sham?
· In line 180, is this referring to a past study? It should be cited. If it is the current study, were subjects aged mice?
Author Response
The author greatly appreciates the feedback of this reviewer and has taken all of the input to heart and in doing the manuscript has been greatly improved. The specific ways the manuscript has been improved in response to the reviewer's suggestions are detailed below.
Major Concerns:
- Introduction/Discussion: The literature has been updated to include more up to date references.
- The first paragraph of the discussion now summarizes the results of the current experiments in more detail.
- The discussion now more directly reflects how the findings of the current paper fit into the recent literature. We have included details about how the current papers fits in relation to the Frye, Lembo and Walf, 2020 paper.
- Details on subject ages (60-120 days), number in each group (n=10), their husbandry e.g. group density (4-5), environment, general care, and IACUC # is now included.
- Statistical methods are now included. One-way and two-way ANOVAs were used for data analyses. The inferential statistics are now included.
- Agreement in tenses has been corrected.
- The figure captions have been updated to be more comprehensive and to include the patterning definitions and the numbers per group are now included.
- Details regarding the cycling and OVX sections have been included. Mice had 7-10 day to recover from OVX. The number of mice that were OVX vs sham are included.
Thank you for your patience with the original manuscript (which clearly was not ready for prime time) and your feedback that helped to improve the product.
Reviewer 2 Report
Comments and Suggestions for Authors
This work does not meet the criteria of a reputable scientific publication and, at the very least, raises my ethical concerns. Right from the start, there are reservations in the form of listed co-authors of the publication, as what does the entry "Dr. Cheryl Anne Frye, Ph.D. and Colleagues" mean, and then there is a mention in the acknowledgment section: "Research herein was supported in part by grants from the National Institute of Mental Health (MH 503 06769801). The content is solely the responsibility of the authors and does not necessarily represent the official views of the National Institutes of Health. Assistance on the studies described herein provided by members of the laboratory, past and present, is greatly appreciated."
There is no approval number from the ethical committee for the experiments conducted in the publication.
As for the text: reading the manuscript, it is difficult to determine whether it is a review or an original work. The introduction does not contain a single reference! Figure 1, included, is unclear if it has ever been published.
There are numerous editorial errors and repetitions throughout. In the experiments section, the number of animals is not provided, and there is no description of the statistics.
Author Response
I appreciate the honest and direct feedback from this reviewer. It must have been difficult to do so but I appreciate the challenging job the reviewer took on.
- This work has been amended so that it may now meet the criteria of a reputable scientific publication.
- The authorship has been amended to remove the statement "and colleagues after" Dr. Cheryl Anne Frye, Ph.D.
- "Research herein was supported in part by grants from the National Institute of Mental Health (MH 503 06769801). The content is solely the responsibility of the authors and does not necessarily represent the official views of the National Institutes of Health. This is a statement that is required by NIH to include when you cite your NIH support.
- Assistance on the studies described herein provided by members of the laboratory, past and present, is greatly appreciated." This statement is included to acknowledge all of the people in my lab who have supported my lab over the years.
- There is now an approval number from the ethical committee for the experiments conducted in the publication, included 08-17.
- The manuscript has been made more easy to read and clearer as an original work.
- The introduction now contains several references. The Figure 1, that was problematic, has been removed.
- The number of animals is now provided
- There is now a description of the statistics.
Thank you for your patience with the original manuscript (which clearly was not ready for prime time) and your feedback that helped to improve the paper.
Reviewer 3 Report
Comments and Suggestions for Authors
- - Title: Needs Revision. It is Unclear.
- - Keywords are not appropriate.
- - Abstract is abrupt, English is not understandable. Abstract needs major revision.
- - Introduction needs revision as line 23-24, line 28 sentence rephrase, line 28 (figure 1, below). There was no need of below or up.
- - Figure 1: not much clear it can further improve.
- - Line 32-33 is incomplete
- - Line 34-37 is confusing
- - Line 42-43 is not clear statement
- - English is very confusing it is not understandable in first read.
- Method: future tense indicates work is to be done.
- - References: very old references are cited. Last reference is of 2020 that shows paper is not updated.
- - In discussion questions are not appealing. There are much formatting errors as underline, italic etc.
- - Many terms and abbreviations are not described and are used frequently throughout the paper.
- - Conclusion is very brief not clearly defines the outcome. Statements are giving incomplete information.
- - Overall very confusing statements which are incomplete also and requires major revision and English correction that may be understandable.
- - Rephrasing of sentences, headings (as many headings of results are same as eg. proestrous c57).
- - Captions of tables and figures needs rephrasing, there are lot of corrections.
- Paper needs major modifications.
Comments on the Quality of English Language
- Overall very confusing statements which are incomplete also and requires major revision and English correction that may be understandable.
Author Response
1..Title: The title has been revised to be clearer.
2. Keywords have been updated.
3. The abstract has been amended.
4. The introduction has been revised.
5. Figure 1 has been removed and replaced by a Diagram which is more contemporary.
6. Method: Future tense has been amended.
7. References: Citations have been updated.
8. Discussion: Question headers have been removed.
9. Many terms and abbreviations are not described and are used frequently throughout the paper.
10. Conclusion has been amended to be more comprehensive.
11. Captions of tables and figures have been rephrased.
12. The paper has had a major overhaul.
Thank you so much for the feedback on this manuscript, which having addressed these comments, has significantly improved the manuscript.
Reviewer 4 Report
Comments and Suggestions for Authors
This MS test the hypothesis that the effects of progesterone on anxiety and depression are not mediated by nuclear PR. The idea is relevant, but it is hard to follow in the text. Actually, I suggest a deep revision of the MS before resubmission, because this version is not ready for peer-reviewing.
Some concerns that were stated when I was trying to read the MS:
The title and the abstract are confusing, and hard to follow. I suggest clarifying the text, presenting the most relevant findings achieved in this study.
The introduction is no sense at all. It is not given the scientific basis for the proposed study. No references are quoted and the authors do not give to the readers the state-of-the-art of the effects of progesterone on anxiety and depression, and the mechanisms of action already reported in the literature.
The authors should explain figure 1, given all the abbreviations in full, and focusing on the most relevant points to readers easily understand the study.
First paragraph in page 2 is incomplete. Then, the second paragraph is repeating the same information as initially presented first, and it is no sense at all. No quotations have been made in this paragraph.
Third paragraph is redundant, and authors lost an opportunity to summarize the most relevant findings about the metabolism of progesterone, and the importance of PR KO animals to elucidate it.
Results should be revised and statistical data (such as F and P values included) into the text.
Figures should be properly insert in the text and should be presented along with a legend.
Methods are not properly described, and must be carefully revised.
The discussion is too long and full of irrelevant information. Please, the text should be revised and focusing on the available literature to explain these data; authors should discuss the limitations and gaps of their findings.
Author Response
I apologize that the manuscipt was not of sufficient quality when originally submitted. It has undergone a deep revision.
1. The title and the abstract have been amended to highlight the most relevant findings achieved in this study.
2. The introduction now gives the readers the state-of-the-art of the effects of progesterone on anxiety and depression, and the mechanisms of action already reported in the literature.
3. Figure 1 has been removed per the suggestion of another reviewer.
4. Results have been revised and statistical data (such as F and P values have been included) into the text.
5. Figures have been properly inserted in the text and are now presented along with legends.
6. Methods are now properly described.
7. The discussion has been amended to focus on the finding and its limitations.
Thank you for your patience with the original manuscript (which clearly was not ready for prime time) and your feedback that helped to improve the manuscript.
Round 2
Reviewer 1 Report
Comments and Suggestions for Authors
Progestogens are hypothesized to have anti-anxiety and anti-depressive effects on behavior, independent of nuclear progestin receptors and dependent on metabolism to allopregnanolone, a metabolite of progesterone.
The manuscript has been greatly improved from the first submission and references have been updated. However, I still find myself checking the references for appropriateness, for example, is “The effect of space travel on human reproductive health” the best reference here? The discussion needs to be shortened and focused.
Of major concern again in the current manuscript is the statistical analyses and results, and without disclosure of the original data, these are impossible to verify. It is strongly encouraged that the author checks the statistics reported for accuracy. For example, in line 203, the F value is 1404.42, which seems high, as do many of the other F-values, given the sample size and ANOVA analysis. Few exact p-values are given with most statistical tests results being reported as P < 0.0001 or P = 0.0001.
· The Figure 1 caption suggests that * = <0.0001 for both panels, yet the report in line 142 suggests P < 0.01 in the forced swim test result.
· The Table 1 caption suggests * = p <0.001 (line 147) yet the report in the text suggests P < 0.0001 (lines 148, 149).
· Figure 6: the y-axis title is covering the units
Minor:
· Table 2 needs adjustments to the spacing of the columns to make it clearer which genotype belongs under which drug
· Line 417 says ‘Proestrous rats…’
Reviewer 3 Report
Comments and Suggestions for Authors
1. Remove Exp 1-9 in abstract.
2. last line of abstract is " In 24 sum, adult female mice can be responsive to P4 for anti-anxiety and anti-depressant behavior, and such effects 25 may be independent of NPRs, but require 5a-reduction, and actions of BDNF in the hippocampus of E2." Please explain in the hippocampus of E2. Is this talking about group receiving E2.
3. Rephrase key words.
4. Methodology: Mention time period of activities performed as elevated plus maze, open field, forced swim test.
5. Remove doubling of headings in method as open field, elevated plus maze, forced swim test.
Comments on the Quality of English Languageminor corrections on english editing required
Reviewer 4 Report
Comments and Suggestions for Authors
The manuscript in my opinion is not confusing, the data description and methods are really poor, the discussion is so long and is not easy to follow.
I listed above some examples of what I mentioned before:
Abstract it is not a fluent text that presents the hypothesis, and consequently exposes the results in a rational manner.
I do not know why the author uses different terms to designate allopregnanolone or proestous (in the abstract and in some moments in the text) without quoting the literature to support it. Actually, I've never listened about these new terms, and if it is quite new, to quote references are very relevant. However, throughout the manuscript the authors use the old and the new terms as their convenience.
The methods are not clear and focused on the details of procedures. A lot of space is spent to discuss about the theoretical of the methods used (and it is not a good/modern and appropriate discussion) and aspects relevant of the procedures were not mentioned, such as the duration of forced swimming test, or an inclusion of a session in the methods describing the drugs (including sources, solubilisation, volume of administration, etc) and treatments schedules, as well a session describing the experimental design (in the abstract 9 experiments were mentioned, but it is clear in the methods), sequence of behavioral tests used and etc.
The results are poorly described, figure 2 and 3 is missing bars legends, figure 6 does not match with the statistical description (in the text we find an output of ANOVA, but it was compared only 2 groups), figures are not representing the individuals (largely used in behavioral studies), it is just the mean +/- error.